# COCO-LM: Correcting and Contrasting Text Sequences for Language Model Pretraining

**Yu Meng**[1]* **Chenyan Xiong**[2] **, Payal Bajaj**[2] **, Saurabh Tiwary**[2] **,**
**Paul Bennett**[2] **, Jiawei Han**[1] **, Xia Song**[2]
[1] University of Illinois at Urbana-Champaign    [2] Microsoft
[1] `{yumeng5,hanj}@illinois.edu`
[2] `{chenyan.xiong,payal.bajaj,satiwary,`
`paul.n.bennett,xiaso}@microsoft.com`

## Abstract

We present a self-supervised learning framework, COCO-LM, that pretrains Language Models by COrrecting and COntrasting corrupted text sequences. Following ELECTRA-style pretraining, COCO-LM employs an auxiliary language model to corrupt text sequences, upon which it constructs two new tasks for pretraining the main model. The first token-level task, Corrective Language Modeling, is to detect and correct tokens replaced by the auxiliary model, in order to better capture token-level semantics. The second sequence-level task, Sequence Contrastive Learning, is to align text sequences originated from the same source input while ensuring uniformity in the representation space. Experiments on GLUE and SQuAD demonstrate that COCO-LM not only outperforms recent state-of-the-art pretrained models in accuracy, but also improves pretraining efficiency. It achieves the MNLI accuracy of ELECTRA with $50\%$ of its pretraining GPU hours. With the same pretraining steps of standard base/large-sized models, COCO-LM outperforms the previous best models by $1+$ GLUE average points.

## 1   Introduction

Pretrained language models (PLMs) have reshaped the way AI systems process natural language [11, 36, 39, 40]. Before task-specific training, it is now a common practice to first pretrain the deep neural networks, often Transformers [53], via a self-supervised token-level language modeling task [29, 31, 40]. Whether it is autoregressive [39], permutational [62], or masked language modeling (MLM) [11], the Transformer networks are pretrained to recover some omitted tokens using the rest of input texts. Then the language semantics captured during pretraining are conveyed to downstream tasks via the pretrained Transformer parameters [5, 8, 44].

Recent research [14, 16, 25, 43] observed several challenges in this self-supervised learning framework. One challenge is its efficiency. After pretrained for a while with the standard token-level language modeling, the networks have already captured the basic language patterns, making a large fraction of pretraining signals no longer informative. Linear improvement in the model effectiveness often requires exponentially more pretraining compute and parameters [25], which is unsustainable. Another challenge is the anisotropy of text representations from pretrained models. The sequence representations from many pretrained models are quite irregular [30, 43] and require dedicated fine-tuning approaches to be useful in sequence-level applications [32, 60].

Clark et al. [7] proposed a new pretraining strategy, ELECTRA, that uses an auxiliary language model ("generator") to replace tokens in input texts and pretrains the main Transformer ("discriminator") to

---

*Part of this work was done while Yu was interning at Microsoft.

35th Conference on Neural Information Processing Systems (NeurIPS 2021).

detect replaced tokens. This improves the pretraining efficiency and effectiveness, but pretraining via binary classification hinders the model's usage on applications requiring language modeling capability (*e.g.*, prompt-based learning [15, 28, 46]). It could further distort the representation space as the Transformers are pretrained to output the same "non-replacement" label for all actual tokens.

In this paper, we present a new self-supervised learning approach, COCO-LM, that pretrains Language Models by COrrecting and COntrasting corrupted text sequences. Following ELECTRA-style pretraining, COCO-LM employs an auxiliary model to corrupt the input texts, upon which it introduces two new pretraining tasks for the main Transformer, one at token level and one at sequence level. The token-level task, corrective language modeling (CLM), pretrains the main Transformer to detect and correct the tokens in the corrupted sequences. It uses a multi-task setup to combine the benefits of replaced token detection and language modeling. The sequence-level task, sequence contrastive learning (SCL), pretrains the model to align text sequences originated from the same source sequence and enforce uniformity of the representation space.

In our experiments on GLUE [54] and SQuAD [41] benchmarks, COCO-LM not only outperforms state-of-the-art pretraining approaches in effectiveness, but also significantly improves the pretraining efficiency. Under the same setting, COCO-LM matches the MNLI accuracy of RoBERTa and ELECTRA with $60\%$ and $50\%$ of their GPU hours in pretraining, respectively. When pretrained with the same number of steps, COCO-LM outperforms the previous best models by $1+$ GLUE average points under the standard base/large-sized model evaluations. With 367 million parameters, COCO-LM$_{\text{Large++}}$ reaches the MNLI accuracy of Megatron$_{3.9\text{B}}$ [49], one of the largest BERT-style model with 3.9 billion parameters. Our analyses provide further insights on the advantage of CLM in learning token representations and its effectiveness in prompted-based fine-tuning, as well as the benefit of SCL in ensuring alignment and uniformity in the representation space for better generalization[1].

## 2   Related Work

Various token-level tasks have been used to pretrain language models. The most classic auto-regressive language modeling is to predict a token given all the previous tokens, or all subsequent ones [36, 39]. BERT uses masked language modeling (MLM) that recovers randomly masked tokens using the rest input. XLNet proposes permutation language modeling that conducts MLM in an autoregressive manner [62]. UniLM uses pseudo MLM which unifies autoregressive and MLM tasks [1, 13].

Sequence-level tasks are also explored, which often pretrain the model to predict certain co-occurrences of sequence pairs. For example, next sentence prediction [11], sentence ordering [27] and previous sentence prediction [56] concatenate two sentences (either correlated or random), and train the Transformer to classify the pair.

Empirically, MLM is still among the most effective tasks to pretrain encoders [29, 31, 40]. RoBERTa [31] found the sentence-level task in BERT not benefitial and discarded it. BART [29] and T5 [40] both observed that MLM is often the most effective task. The empirical advantages of other pretraining tasks are more task-specific, for example, entity related masks for knowledge intensive applications [20, 24], and sequence-level tasks for long form text modeling [42].

Instead of randomly altering texts, ELECTRA [7] uses a smaller auxiliary Transformer pretrained by MLM to replace some tokens in the text sequences using its language modeling probability, and pretrains the main Transformer to detect the replaced tokens. ELECTRA achieves state-of-the-art accuracy in many language tasks [7]. Later, Clark et el. [6] developed ELECTRIC, which pretrains encoders by contrasting original tokens against negatives sampled from a cloze model. ELECTRIC re-enables the language modeling capability but underperforms ELECTRA in downstream tasks.

Our work is also related to contrastive learning which has shown great success in visual representation learning [4, 22, 34]. Its effectiveness of in language is more observed in the fine-tuning stage, for example, in sentence representation [16], dense retrieval [60], and GLUE fine-tuning [19].

## 3   Method

We present the preliminaries of PLMs, their challenges, and the new COCO-LM framework.

---

[1]Code and pretrained models can be found at `https://github.com/microsoft/COCO-LM`.

## 3.1 Preliminary on Language Model Pretraining

In this work we focus on pretraining BERT-style bidirectional Transformer encoders [11] that are widely used in language representation tasks. We first recap the masked language modeling (MLM) task introduced by BERT [11] and then discuss the pretraining framework of ELECTRA [7].

**BERT Pretraining** uses the masked language modeling task (MLM) [11], which is to take an input sequence $X^{\text{orig}} = [x_1^{\text{orig}}, \ldots, x_i^{\text{orig}}, \ldots, x_n^{\text{orig}}]$, with 15% random tokens replaced by [MASK] symbols (*e.g.*, the $i$-th token), and train the model to predict the original tokens at the masked positions:

$$\left[x_1^{\text{orig}}, \ldots, [\texttt{MASK}]_i, \ldots, x_n^{\text{orig}}\right] \xrightarrow{\text{Transformer}} \boldsymbol{H} \xrightarrow{\text{MLM Head}} p_{\text{MLM}}(x|\boldsymbol{h}_i),$$

where the Transformer generates contextualized representations $\boldsymbol{H} = \{\boldsymbol{h}_i\}_{i=1}^n$. The MLM Head predicts the masked token from the vocabulary $V$ using the hidden representation $\boldsymbol{h}_i$ and token embeddings $\boldsymbol{x}$. The pretraining minimizes the MLM loss on the set of masked positions $\mathcal{M}$. Specifically,

$$p_{\text{MLM}}(x|\boldsymbol{h}_i) = \frac{\exp(\boldsymbol{x}^\top \boldsymbol{h}_i)}{\sum_{x_t \in V} \exp(\boldsymbol{x}_t^\top \boldsymbol{h}_i)}; \quad \mathcal{L}_{\text{MLM}} = \mathbb{E}\left(-\sum_{i \in \mathcal{M}} \log p_{\text{MLM}}\left(x_i^{\text{orig}}|\boldsymbol{h}_i\right)\right).$$

**ELECTRA Pretraining** uses two Transformers, a "generator" pretrained by MLM, and a "discriminator" pretrained using the generator's outputs. We refer them as *auxiliary* and *main* Transformers, as the former is discarded after pretraining and the latter may be trained by "generative" tasks too.

The auxiliary model outputs a corrupted sequence $X^{\text{MLM}}$ by sampling from its predicted probability:

$$x_i^{\text{MLM}} \sim p_{\text{MLM}}\left(x|\boldsymbol{h}_i\right), \text{ if } i \in \mathcal{M}; \quad x_i^{\text{MLM}} = x_i^{\text{orig}}, \text{ else.} \tag{1}$$

The masked positions are replaced by sampled tokens considered plausible in context by the auxiliary Transformer, which are more deceiving than random replacements. ELECTRA uses a skinnier auxiliary network (*e.g.*, hidden dimension is $1/3$ of the main model) to control the signal difficulty.

The main Transformer takes $X^{\text{MLM}}$ and classifies the replaced tokens:

$$X^{\text{MLM}} \xrightarrow{\text{Main Transformer}} \boldsymbol{H} \xrightarrow{\text{RTD Head}} p_{\text{RTD}}\left(\mathbb{1}(x_i^{\text{MLM}} = x_i^{\text{orig}})|\boldsymbol{h}_i\right),$$

where $\mathbb{1}(\cdot)$ is the indicator function. The Replaced Token Detection (RTD) head uses a sigmoid linear layer to output the binary probability, and the main Transformer is trained with binary cross entropy loss. The RTD task is trained on all tokens instead of masked ones and improves efficiency.

The two Transformers are pretrained jointly. The auxiliary model gradually generates more realistic replacement tokens and the main model learns to better detect them. This forms a natural learning curriculum and significantly improves ELECTRA's accuracy in downstream tasks [7].

## 3.2 Challenges of ELECTRA-Style Pretraining

**Missing Language Modeling Benefits.** The classification task in ELECTRA is simpler and more stable [61], but raises two challenges. The first is the lack of language modeling capability which is a necessity in some tasks [6]. For example, prompt-based learning requires a language model to generate labels [15, 33, 45, 46]. The second is that the binary classification task may not be sufficient to capture certain word-level semantics that are critical for token-level tasks.

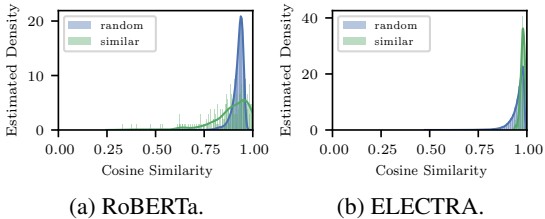

(a) RoBERTa.      (b) ELECTRA.

Figure 1: Cosine similarity distributions of random/similar sequence pairs using [CLS] embeddings from pretrained models. Histograms/curves are distribution bins/kernel density estimates.

**Squeezing Representation Space.** Another challenge is that the representations from Transformer-based language models often reside in a narrow cone, where two random sentences have high similarity scores (lack of uniformity), and closely related sentences may have more different representations (lack of alignment) [14, 16, 30].

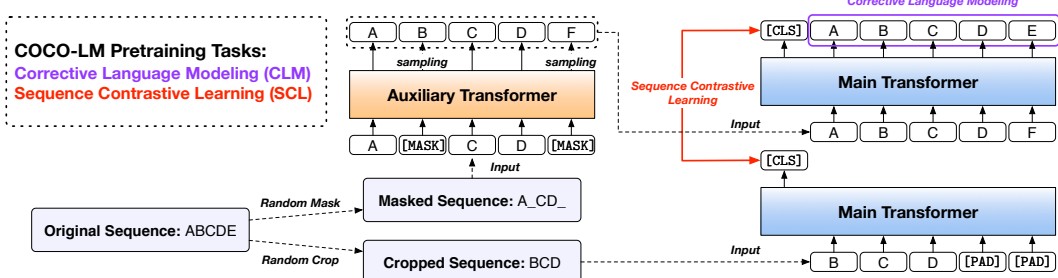

Figure 2: The overview of COCO-LM. The auxiliary Transformer is pretrained by MLM. Its corrupted text sequence is used as the main Transformer's pretraining input in Corrective Language Modeling and paired with the cropped original sequence for Sequence Contrastive Learning.

Figure 1 illustrates such behaviors with random sentence pairs (from pretraining corpus) and semantically similar pairs (those annotated with maximum similarity from STS-B [3]). With RoBERTa, the cosine similarities of most random sentence pairs are near $0.8$, bigger than many semantically similar pairs. The representation space from ELECTRA is even more squeezed. Nearly all sentence pairs, both random and similar ones, have around $0.9$ cosine similarity. This may not be surprising as ELECTRA is pretrained to predict the same output ("non-replacement") for all tokens in these sequences. The irregular representation space raises the risk of degeneration [37, 55] and often necessitates sophisticated post-adjustment or fine-tuning to improve the sequence representations [16, 30, 32, 60].

### 3.3 COCO-LM Pretraining

COCO-LM also employs an auxiliary Transformer to construct the corrupted text sequence, as in Eqn. (1), but it introduces two new pretraining tasks upon the corrupted sequences to address the challenges previously described. In the rest of this section, we present these two tasks and then the detailed configurations of COCO-LM. Its framework is illustrated in Figure 2.

**Corrective Language Modeling** (CLM) trains the main Transformer to recover the original tokens, given the corrupted text sequence $X^{\text{MLM}}$:

$$X^{\text{MLM}} \xrightarrow{\text{Main Transformer}} \boldsymbol{H} \xrightarrow{\text{CLM Head}} p_{\text{CLM}}(x|\boldsymbol{h}_i).$$

The CLM Head uses the hidden representations $\boldsymbol{H}$ to output a language modeling probability, instead of a binary classification score. The forward pass of the CLM Head is the same as All-Token MLM, a variation of ELECTRA [7] that consists of a language modeling layer and a binary classification layer for the copy mechanism:

$$p_{\text{LM}}(x_i|\boldsymbol{h}_i) = \mathbb{1}\left(x_i = x_i^{\text{MLM}}\right) p_{\text{copy}}(1|\boldsymbol{h}_i) + p_{\text{copy}}(0|\boldsymbol{h}_i) \frac{\exp(\boldsymbol{x}_i^{\top}\boldsymbol{h}_i)}{\sum_{x_t \in V} \exp(\boldsymbol{x}_t^{\top}\boldsymbol{h}_i)},$$

$$p_{\text{copy}}(y_i|\boldsymbol{h}_i) = \exp(y_i \cdot \boldsymbol{w}_{\text{copy}}^{\top}\boldsymbol{h}_i)/\left(\exp(\boldsymbol{w}_{\text{copy}}^{\top}\boldsymbol{h}_i) + 1\right),$$

where $\boldsymbol{w}_{\text{copy}}$ is a learnable weight and $p_{\text{copy}}(y_i|\boldsymbol{h}_i)$ is the copy mechanism ($y_i = 1$ when the input token is original and can be directly copied to the output; $y_i = 0$ when the input token needs to be corrected to another token from the vocabulary).

In ELECTRA, All-Token MLM performs worse than RTD [7]. Language modeling on the corrupted text sequence $X^{\text{MLM}}$ is hard as the replaced tokens from the auxiliary model are more deceiving than [MASK]. To improve the language model learning, different from All-Token MLM, CLM employs a

multi-task setup that combines the RTD task to explicitly train the copy mechanism $p_{\text{copy}}(\cdot)$:

$$\mathcal{L}_{\text{copy}} = -\mathbb{E}\left(\sum_{i=1}^{n} \mathbb{1}\left(x_i^{\text{MLM}} = x_i^{\text{orig}}\right) \log p_{\text{copy}}(1|\boldsymbol{h}_i) + \mathbb{1}\left(x_i^{\text{MLM}} \neq x_i^{\text{orig}}\right) \log p_{\text{copy}}(0|\boldsymbol{h}_i)\right), \quad (2)$$

$$\mathcal{L}_{\text{LM}} = -\mathbb{E}\left(\sum_{i\in\mathcal{M}} \log p_{\text{LM}}\left(x_i^{\text{orig}}|\boldsymbol{h}_i\right)\right)$$

$$= -\mathbb{E}\left(\sum_{i\in\mathcal{M}} \log\left(\mathbb{1}\left(x_i^{\text{MLM}} = x_i^{\text{orig}}\right) p_{\text{copy}}^{\text{sg}}(1|\boldsymbol{h}_i) + p_{\text{copy}}^{\text{sg}}(0|\boldsymbol{h}_i) \frac{\exp(\boldsymbol{x}_i^{\top}\boldsymbol{h}_i)}{\sum_{x_t \in V}\exp(\boldsymbol{x}_t^{\top}\boldsymbol{h}_i)}\right)\right),$$

$$\mathcal{L}_{\text{CLM}} = \lambda_{\text{copy}}\mathcal{L}_{\text{copy}} + \mathcal{L}_{\text{LM}}.$$

The hyperparameter $\lambda_{\text{copy}}$ balances the weights of the two tasks. The binary cross entropy loss in Eqn. (2) explicitly trains the copy probability. We also use stop gradient (sg) to decouple the gradient backpropagation to $p_{\text{copy}}(\cdot)$ from the LM task. This way, the main Transformer first learns the easier classification task and then uses it to help learn the harder LM task. The binary classification task is trained on all tokens while the language modeling task is trained only on masked positions.

CLM combines the advantages of MLM and ELECTRA: The main Transformer is trained on all tokens with the help of the binary classification task while also being able to predict words, thus enjoying the efficiency benefits of ELECTRA and preserving the language modeling benefits.

**Sequence Contrastive Learning** (SCL) forms a contrastive learning objective upon the sequence embeddings to learn more robust representations. Broadly, contrastive learning is to align a positive pair of instances, often different views of the same information [4, 34], in contrast to unrelated negative instances [22, 60]. The different views are often obtained by applying data augmentations on the same input, for example, rotation, cropping, and blurring on visual representations [4, 34], so that the neural networks can learn representations robust to these data alterations.

In COCO-LM, the corrupted sequence $X^{\text{MLM}}$ already provides a form of data augmentation. We pair it with another augmentation, $X^{\text{crop}}$, a randomly cropped contiguous span of $X^{\text{orig}}$ (the length of $X^{\text{crop}}$ is 90% of $X^{\text{orig}}$ so that the major sequence meaning is preserved), to construct the positive pair and to contrast with random negatives.

Specifically, a training batch $B$ in SCL includes a random set of corrupted and cropped sequences: $B = \{(X_1^{\text{MLM}}, X_1^{\text{crop}}), \ldots, (X_N^{\text{MLM}}, X_N^{\text{crop}})\}$, with $X_k^{\text{MLM}}$ and $X_k^{\text{crop}}$ originated from $X_k^{\text{orig}}$. A positive contrastive pair $(X, X^+)$ consists of either $(X_k^{\text{MLM}}, X_k^{\text{crop}})$ or $(X_k^{\text{crop}}, X_k^{\text{MLM}})$ (symmetrical contrast). The negative instances are all the remaining sequences in the batch $B^- = B \setminus \{(X, X^+)\}$. The contrastive loss is formulated as:

$$\mathcal{L}_{\text{SCL}} = -\mathbb{E}\left(\log \frac{\exp(\cos(\boldsymbol{s}, \boldsymbol{s}^+)/\tau)}{\exp(\cos(\boldsymbol{s}, \boldsymbol{s}^+)/\tau) + \sum_{X^- \in B^-}\exp(\cos(\boldsymbol{s}, \boldsymbol{s}^-)/\tau)}\right),$$

$$= -\mathbb{E}\left(\cos(\boldsymbol{s}, \boldsymbol{s}^+)/\tau - \log\left(\exp(\cos(\boldsymbol{s}, \boldsymbol{s}^+)/\tau) + \sum_{X^- \in B^-}\exp\left(\cos(\boldsymbol{s}, \boldsymbol{s}^-)/\tau\right)\right)\right), \quad (3)$$

where $\boldsymbol{s}, \boldsymbol{s}^+, \boldsymbol{s}^-$ are the representations of $X, X^+, X^-$, respectively, from the main Transformer (*i.e.*, $\boldsymbol{h}_{\text{[CLS]}}$). The similarity metric is cosine similarity (cos) and the temperature $\tau$ is set to 1.

As shown in Wang et al. [55], the first term in Eqn. (3) ($\cos(\boldsymbol{s}, \boldsymbol{s}^+)$) improves *alignment* of the space. It encourages representations to be robust to the corruptions and the alterations on the original text. The second term in Eqn. (3) promotes *uniformity*. It pushes unrelated sequences apart in the representation space and ensures low cosine similarity between random data points. Several studies have observed improved generalization ability from better alignment and uniformity [16, 37, 55].

Aligning $X^{\text{MLM}}$ with $X^{\text{crop}}$ requires the main Transformer to produce sequence representations robust to both token-level (*i.e.*, MLM replacements) and sequence-level (*i.e.*, cropping) alterations. The model is thus encouraged to reason more using partially altered sequences to recover the original information.

**Overall Training.** COCO-LM uses the following loss function:

$$\mathcal{L}_{\text{COCO-LM}} = \mathcal{L}_{\text{MLM}}^{\text{Aux.}} + \mathcal{L}_{\text{CLM}}^{\text{Main}} + \mathcal{L}_{\text{SCL}}^{\text{Main}}. \tag{4}$$

The auxiliary Transformer is pretrained by masked language modeling (MLM) and generates corrupted sequences. The main Transformer is pretrained to correct the corruption (CLM) and to contrast the corrupted sequences with the cropped sequences (SCL). The two Transformers are pretrained jointly with the loss in Eqn. (4). The main Transformer is used in downstream applications.

**Network Configurations.** Similar to ELECTRA, the auxiliary Transformer is smaller than the main model, but we use different configurations in the auxiliary model: (1) We reduce the number of layers to $1/3$ or $1/4$ (under *base* or *large* model setup, respectively) but keep its hidden dimension the same with the main model, instead of shrinking its hidden dimensions; (2) We disable dropout in it when sampling replacement tokens. We find such configurations empirically more effective and use them as the backbone of COCO-LM. The main Transformer follows the standard architecture of BERT/ELECTRA and can be easily adopted by downstream application pipelines with almost no changes.

## 4    Experimental Setup

**Pretraining Settings.** We employ three standard settings, *base*, *base++*, and *large++*. *Base* is the BERT$_{\text{Base}}$ training configuration [11]: Pretraining on Wikipedia and BookCorpus [63] (16 GB of texts) for 256 million samples on 512 token sequences (125K batches with 2048 batch size). We use the same corpus and $32,768$ uncased BPE vocabulary [47] as with TUPE [26].

*Base++* trains the base size model with larger corpora and/or more training steps. Following recent research [1, 31, 62], we add in OpenWebText [18], CC-News [31], and STORIES [52], to a total of 160 GB texts, and train for 4 billion (with 2048 batch size) samples [31]. We follow the prepossessing of UniLMV2 [1] and use $64,000$ cased BPE vocabulary.

*Large++* uses the same training corpora as *base++* and pretrains for 4 billion samples (2048 batch size). Its Transformer configuration is the same with BERT$_{\text{Large}}$ [11].

**Model Architecture.** Our *base/base++* model uses the BERT$_{\text{Base}}$ architecture [11]: 12 layer Transformer, 768 hidden size, plus T5 relative position encoding [40]. Our *large++* model is the same with BERT$_{\text{Large}}$, 24 layer and 1024 hidden size, plus T5 relative position encoding [40]. Our auxiliary network uses the same hidden size but a shallow 4-layer Transformer in *base/base++* and a 6-layer one in *large++*. When generating $X^{\text{MLM}}$ we disable dropout in the auxiliary model.

**Downstream Tasks.** We use the tasks included in GLUE [54] and SQuAD 2.0 reading compression [41]. Please refer to Appendix A for more details about GLUE tasks. Standard hyperparameter search in fine-tuning is performed, and the search space can be found in Appendix B. The fine-tuning protocols use the open-source implementation of TUPE [26]. The reported results are the median of five random seeds on GLUE and SQuAD.

**Baselines.** We compare with various pretrained models in each setting. To reduce the variance in data processing/environments, we also pretrain and fine-tune RoBERTa and ELECTRA under exactly the same setting with COCO-LM, marked with "(Ours)". All numbers unless marked by "(Ours)" are from reported results in recent research (more details in Appendix C).

**Implementation Details.** Our implementation builds upon the open-source implementation from MC-BERT [61] and fairseq [35]. More implementation details are mentioned in Appendix D.

## 5    Evaluation Results

Three groups of experiments are conducted to evaluate COCO-LM and its two new pretraining tasks.

### 5.1    Overall Results and Ablations

**Overall Results** are listed in Table 1. Under all three settings, COCO-LM outperforms all recent state-of-the-art pretraining models on GLUE average and SQuAD. It improves the state-of-the-art GLUE score by about one point under all three settings. COCO-LM also enjoys better parameter efficiency. Using less than $10\%$ of Megatron's parameters, COCO-LM$_{\text{Large++}}$ matches the MNLI accuracy of Megatron$_{3.9\text{B}}$, one of the largest pretrained BERT-style encoders.

| Model | Params | GLUE Single Task | | | | | | | | | SQuAD 2.0 | |
|---|---|---|---|---|---|---|---|---|---|---|---|---|
| | | MNLI-(m/mm) | QQP | QNLI | SST-2 | CoLA | RTE | MRPC | STS-B | AVG | EM | F1 |
| **Base Setting:** BERT Base Size, Wikipedia + Book Corpus (16GB) | | | | | | | | | | | | |
| BERT [11] | 110M | 84.5/- | 91.3 | 91.7 | 93.2 | 58.9 | 68.6 | 87.3 | 89.5 | 83.1 | 73.7 | 76.3 |
| RoBERTa [31] | 125M | 84.7/- | – | – | 92.7 | – | – | – | – | – | – | 79.7 |
| XLNet [62] | 110M | 85.8/85.4 | – | – | 92.7 | – | – | – | – | – | 78.5 | 81.3 |
| ELECTRA [7] | 110M | 86.0/85.3 | 90.0 | 91.9 | 93.4 | 64.3 | 70.8 | 84.9 | 89.1 | 83.7 | 80.5 | 83.3 |
| MC-BERT [61] | 110M | 85.7/85.2 | 89.7 | 91.3 | 92.3 | 62.1 | 75.0 | 86.0 | 88.0 | 83.7 | – | – |
| DeBERTa [23] | 134M | 86.3/86.2 | – | – | – | – | – | – | – | – | 79.3 | 82.5 |
| TUPE [26] | 110M | 86.2/86.2 | 91.3 | 92.2 | 93.3 | 63.6 | 73.6 | 89.9 | 89.2 | 84.9 | – | – |
| RoBERTa (Ours) | 110M | 85.8/85.5 | 91.3 | 92.0 | **93.7** | 60.1 | 68.2 | 87.3 | 88.5 | 83.3 | 77.7 | 80.5 |
| ELECTRA (Ours) | 110M | 86.9/86.7 | 91.9 | 92.6 | 93.6 | **66.2** | 75.1 | 88.2 | 89.7 | 85.5 | 79.7 | 82.6 |
| COCO-LM | 110M | **88.5/88.3** | **92.0** | **93.1** | 93.2 | 63.9 | **84.8** | **91.4** | **90.3** | **87.2** | **82.4** | **85.2** |
| **Base++ Setting:** BERT Base Size, Bigger Training Data, and/or More Training Steps | | | | | | | | | | | | |
| XLNet [62] | 110M | 86.8/- | 91.4 | 91.7 | 94.7 | 60.2 | 74.0 | 88.2 | 89.5 | 84.6 | 80.2 | – |
| RoBERTa [31] | 125M | 87.6/- | 91.9 | 92.8 | 94.8 | 63.6 | 78.7 | 90.2 | 91.2 | 86.4 | 80.5 | 83.7 |
| UniLM V2 [1] | 110M | 88.5/- | 91.7 | 93.5 | **95.1** | 65.2 | 81.3 | **91.8** | 91.0 | 87.1 | 83.3 | 86.1 |
| DeBERTa [23] | 134M | 88.8/88.5 | – | – | – | – | – | – | – | – | 83.1 | 86.2 |
| CLEAR [59] | 110M | 86.7/- | 90.0 | 92.9 | 94.5 | 64.3 | 78.3 | 89.2 | 89.8 | 85.7 | – | – |
| COCO-LM | 134M | **90.2/90.0** | **92.2** | **94.2** | 94.6 | **67.3** | **87.4** | 91.2 | **91.8** | **88.6** | **85.4** | **88.1** |
| **Large++ Setting:** BERT Large Size, Bigger Training Data, and More Training Steps | | | | | | | | | | | | |
| XLNet [62] | 360M | 90.8/90.8 | 92.3 | 94.9 | **97.0** | 69.0 | 85.9 | 90.8 | 92.5 | 89.2 | 87.9 | 90.6 |
| RoBERTa [31] | 356M | 90.2/90.2 | 92.2 | 94.7 | 96.4 | 68.0 | 86.6 | 90.9 | 92.4 | 88.9 | 86.5 | 89.4 |
| ELECTRA [7] | 335M | 90.9/- | 92.4 | 95.0 | 96.9 | 69.1 | 88.0 | 90.8 | 92.6 | 89.4 | 88.0 | 90.6 |
| DeBERTa [23] | 384M | 91.1/91.1 | 92.3 | 95.3 | 96.8 | 70.5 | – | – | – | – | 88.0 | 90.7 |
| COCO-LM | 367M | **91.4/91.6** | **92.8** | **95.7** | 96.9 | **73.9** | **91.0** | **92.2** | **92.7** | **90.8** | **88.2** | **91.0** |
| Megatron_{1.3B} [49] | 1.3B | 90.9/91.0 | 92.6 | – | – | – | – | – | – | – | 87.1 | 90.2 |
| Megatron_{3.9B} [49] | 3.9B | 91.4/91.4 | 92.7 | – | – | – | – | – | – | – | 88.5 | 91.2 |

Table 1: Results on GLUE and SQuAD 2.0 development set. All results are single-task, single-model fine-tuning. Results not available in public reports are marked as "–". DeBERTa reported RTE, MRPC and STS-B results by fine-tuning from MNLI checkpoints which are not single-task results. We use Spearman correlation for STS, Matthews correlation for CoLA, and accuracy for the rest on GLUE. AVG is the average of the eight tasks on GLUE. All baseline results unless marked by (Ours) are reported by previous research.

| Model | Params | MNLI-(m/mm) | QQP | QNLI | SST-2 | CoLA | RTE | MRPC | STS-B | AVG |
|---|---|---|---|---|---|---|---|---|---|---|
| **Base/Base++ Setting:** BERT Base Size | | | | | | | | | | |
| BERT_{Base} | 110M | 84.6/83.4 | 89.2 | 90.5 | 93.5 | 52.1 | 66.4 | 84.8 | 85.8 | 80.8 |
| ELECTRA_{Base++} | 110M | 88.5/88.0 | 89.5 | 93.1 | **96.0** | 64.6 | 75.2 | 88.1 | 90.2 | 85.6 |
| COCO-LM_{Base++} | 134M | **89.8/89.3** | **89.8** | **94.2** | 95.6 | **68.6** | **82.3** | **88.5** | **90.3** | **87.4** |
| **Large/Large++ Setting:** BERT Large Size | | | | | | | | | | |
| BERT_{Large} | 335M | 86.7/85.9 | 89.3 | 92.7 | 94.9 | 60.5 | 70.1 | 85.4 | 86.5 | 83.2 |
| ELECTRA_{Large++} | 335M | 90.7/90.2 | 90.4 | 95.5 | **96.7** | 68.1 | 86.1 | **89.2** | 91.7 | 88.5 |
| COCO-LM_{Large++} | 367M | **91.6/91.1** | 90.5 | 95.8 | **96.7** | 70.5 | 89.2 | 88.4 | **91.8** | **89.3** |

Table 2: GLUE test set results obtained from the GLUE leaderboard. We perform hyperparameter search for each task with ten random seeds and use the best development set model for test predictions. All results are from vanilla single-task fine-tuning (no ensemble, task-specific tricks, etc.).

Table 2 shows GLUE test set results which further confirm the advantages of COCO-LM over previous methods.

**Efficiency.** In downstream tasks, the efficiency of COCO-LM is the same with BERT. In pretraining, the auxiliary model and SCL introduce extra cost. However, as shown in Figure 3, COCO-LM is more efficient in GPU hours. It outperforms RoBERTa & ELECTRA by 1+ points on MNLI with the same GPU hours and reaches their accuracy with around 60% & 50% GPU hours, respectively.

**Ablation Studies.** Table 3 shows the ablations of COCO-LM under the *base* setting on GLUE DEV.

*Pretraining Task.* With only RTD, our backbone model with the shallow auxiliary Transformer is quite effective. CLM and SCL both provide additional improvements on MNLI and GLUE average. Their advantages are better observed on different tasks, for example, CLM on MNLI-mm and SCL on RTE and MRPC. Combining the two in COCO-LM provides better overall effectiveness. In later experiments, we further analyze the benefits of these two tasks.

| Group | Method | MNLI-(m/mm) | QQP | QNLI | SST-2 | CoLA | RTE | MRPC | STS-B | AVG |
|---|---|---|---|---|---|---|---|---|---|---|
| | COCO-LM$_{\text{Base}}$ | 88.5/88.3 | 92.0 | 93.1 | 93.2 | 63.9 | 84.8 | 91.4 | 90.3 | 87.2 |
| **Pretraining Task** | RTD Only | 88.4/88.2 | 92.1 | 93.5 | 92.7 | 67.3 | 80.5 | 89.0 | 90.9 | 86.8 |
| | CLM Only | 88.6/88.4 | 92.0 | 93.2 | 93.7 | 67.4 | 80.1 | 90.0 | 90.4 | 86.9 |
| | SCL + RTD | 88.6/88.2 | 92.1 | 93.5 | 93.8 | 64.3 | 82.7 | 90.2 | 90.6 | 86.9 |
| **Network Setting** | w/o. Rel-Pos | 88.2/87.7 | 92.2 | 93.4 | 93.7 | 68.8 | 82.7 | 91.2 | 90.6 | 87.6 |
| | w. ELECTRA's Auxiliary | 88.0/87.7 | 91.9 | 92.7 | 93.5 | 64.3 | 81.2 | 89.5 | 89.7 | 86.3 |
| **Training Signal** | w. Random Replacements | 84.9/84.7 | 91.4 | 91.1 | 91.4 | 41.6 | 70.0 | 87.3 | 87.1 | 80.6 |
| | w. Converged Auxiliary | 88.3/88.1 | 92.0 | 92.8 | 94.3 | 64.2 | 78.3 | 90.4 | 90.2 | 86.3 |
| **CLM Setup** | All-Token LM Only | 87.2/87.0 | 91.8 | 92.6 | 93.7 | 60.6 | 74.0 | 88.5 | 89.7 | 84.7 |
| | CLM w/o. Copy | 88.0/87.9 | 91.8 | 93.1 | 94.4 | 66.6 | 76.9 | 89.5 | 90.1 | 86.3 |
| | CLM w/o. Stop-grad | 88.5/88.2 | 92.0 | 92.9 | 94.3 | 66.5 | 80.9 | 90.0 | 90.6 | 86.9 |

Table 3: Ablations on GLUE Dev. that eliminate (w/o.), keep (Only) or switch (w.) one component.

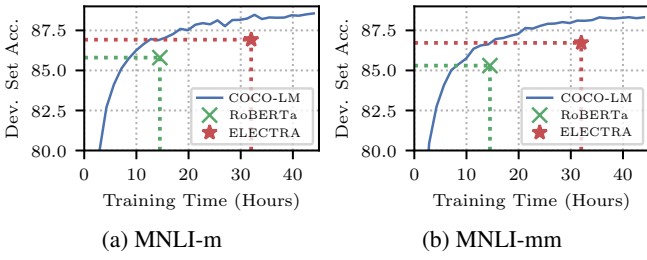

(a) MNLI-m  (b) MNLI-mm

Figure 3: COCO-LM$_{\text{Base}}$ on MNLI Dev. ($y$-axes) at different pretraining hours on four DGX-2 nodes (64 V100 GPUs). The final training hours and accuracy of RoBERTa (Ours) and ELECTRA (Ours) measured in the same settings are marked.

Figure 4: The performance of COCO-LM$_{\text{Base}}$ when pretrained with different crop fractions. The $x$-axis is the fraction of $X^{\text{orig}}$ being kept (no cropping is $100\%$).

*Architecture.* Removing relative position encoding (Rel-Pos) leads to better numbers on some tasks but significantly hurts MNLI. Using a shallow auxiliary network and keeping the same hidden dimension (768) is more effective than ELECTRA's 12-layer but 256-hidden dimension generator.

*Pretraining Signal Construction.* Using randomly replaced tokens to corrupt text sequence hurts significantly. Using a converged auxiliary network to pretrain the main model also hurts. It is better to pretrain the two Transformers together, as the auxiliary model gradually increases the difficulty of the corrupted sequences and provides a natural learning curriculum for the main Transformer.

*CLM Setup.* Disabling the multi-task learning and using All-Token MLM [7] reduces model accuracy. The copy mechanism is effective. The benefits of the stop gradient operation are more on stability (preventing training divergence).

## 5.2 Analyses of Contrastive Learning with SCL

This group of experiments analyzes the behavior of SCL. All experiments use the *base* setting.

**Ablation on Data Augmentation.** Figure 4 shows the effects of the cropping operation when forming positive SCL pairs with the corrupted sequence. Using the original sequence results in worse GLUE accuracy. It is less informative as the model no longer needs to learn representations robust to sequence-level alteration. Cropping too much (*e.g.*, only keeping $70\%$ of the original sequence), may hurt as it can alter the semantics too much. Empirically a simple alteration works the best, similar to the observations in recent research [4, 16, 22].

**Alignment and Uniformity.** Figure 5 plots the distribution of cosine similarities between random sequence pairs and similar ones using representations pretrained by COCO-LM. The representation space from COCO-LM is drastically different from those in Figure 1. With COCO-LM, similar pairs are more *aligned* and random pairs are distributed more *uniformly*. Many similar pairs have near 1 cosine similarity and are clearly separated from random pairs which center around 0. The t-SNE [9] plot in Figure 6 further demonstrates the benefits of SCL. The similar sentence pairs (marked by same shapes) are *aligned* closer when pretrained with SCL. Their average cosine similarity is $0.925$ when

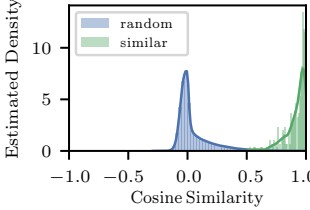
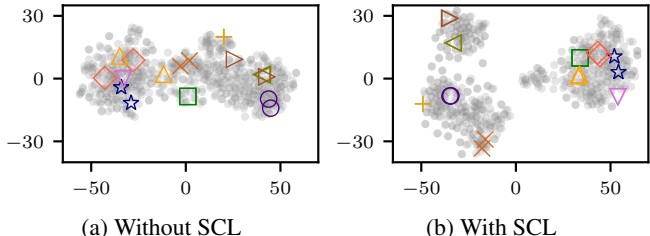

(a) Without SCL       (b) With SCL

Figure 5: Cosine similarity of sequence pairs randomly sampled from pretraining corpus and most similar pairs from STS-B using `[CLS]` from COCO-LM$_{\text{Base}}$.

Figure 6: The t-SNE of sequence representations learned with or without SCL. The points are sampled from the most semantically similar sentences pairs from STS-B (with 5-score labels). The `[CLS]` embeddings are not fine-tuned. Some randomly selected similar pairs are marked by same shapes.

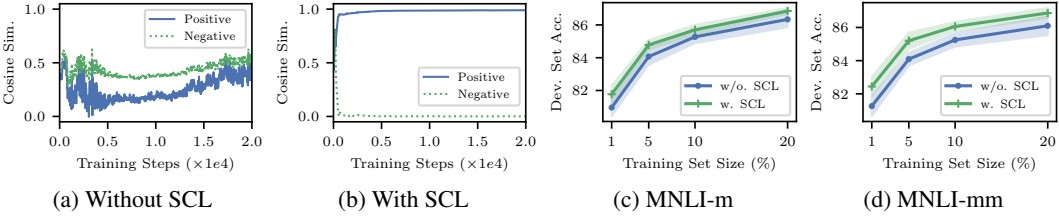

(a) Without SCL     (b) With SCL     (c) MNLI-m     (d) MNLI-mm

Figure 7: Analyses of SCL. Figs. (a) and (b) show the average cosine similarity between the `[CLS]` embeddings of positive and negative contrastive pairs during pretraining. Figs. (c) and (d) show the few-shot accuracy on MNLI with different fractions of MNLI training set used ($x$-axes). The error bars mark the max/min and the solid lines are the average of five fine-tuning runs.

pretrained with SCL, while is $0.863$ without SCL. This better alignment and uniformity is achieved by COCO-LM with SCL via pretraining, without using task-specific data nor supervised labels.

**Regularizing the Representation Learning for Better Few-Shot Ability.** One would expect any pretrained Transformers to easily align a pair of corrupted sequence and cropped sequence as the two share about $80\%$ tokens. However, as shown in Figure 7a, that is not the case: Without SCL, the cosine similarity of the positive pairs is even lower than random negatives. SCL is necessary to regularize the representation space and to reduce the risk of degeneration (Figure 7b).

Similar to empirical observations and theoretical analyses in recent research [14, 16, 55], a more regularized representation space results in better generalization ability in scenarios with limited labels. Figure 7c and 7d show the results when COCO-LM are trained (via standard fine-tuning) with only a fraction of MNLI labels. The improvements brought by SCL are more significant when fewer fine-tuning labels are available. With $1\%$ MNLI labels, pretraining with SCL improves MNLI-m/mm accuracy by $0.8/0.5$ compared to that without SCL. Using only $10\%/20\%$ labels, COCO-LM with SCL reaches similar MNLI accuracy with RoBERTa (Ours)/ELECTRA (Ours) fine-tuned with all labels, respectively.

## 5.3 Analyses of Language Modeling with CLM

The last group of experiments studies the effectiveness and benefits of CLM.

**Ablations on Training Configurations.** Figure 8 illustrates pretraining process with CLM and All-Token MLM. The plots demonstrate the difficulty of language modeling upon corrupted text sequences. It is quite an unbalanced task. For the majority of the tokens (Original) the task is simply to copy its input at the same position. For the replaced tokens ($7 - 8\%$ total), however, the model needs to detect the abnormality brought by the auxiliary model and recover the original token. Implicitly training the copy mechanism as part of the hard LM task is not effective: The copy accuracy of All-Token MLM is much lower, and thus the LM head may confuse original tokens with replaced ones. As shown in Table 3 and ELECTRA [7], pretraining with All-Token MLM performs worse than using the RTD task, though the latter is equivalent to only training the copy mechanism. The multi-task learning of CLM is necessary for the main Transformer to stably learn the language modeling task upon the corrupted text sequence.

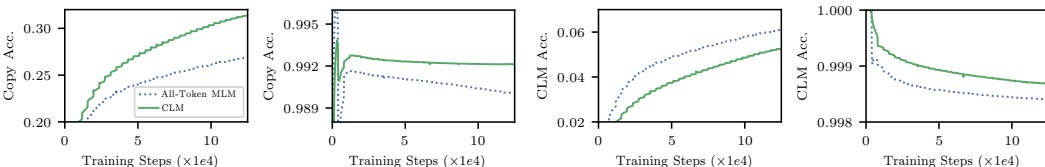

(a) Copy Acc. (Replaced)  (b) Copy Acc. (Original)  (c) CLM Acc. (Replaced)  (d) CLM Acc. (Original)

Figure 8: The copying accuracy and the language modeling accuracy ($y$-axes) of CLM and All-Token MLM at different pretraining steps ($x$-axes, in 10K scale). The accuracy is averaged on tokens that are replaced by the auxiliary Transformer (Replaced) or those from the original input text (Original).

**Prompt-Based Fine-Tuning with CLM.** Table 4 includes the prompt-based fine-tuning experiments on MNLI for RoBERTa and COCO-LM under *base++* and *large++* sizes, following the same few-shot manual prompt fine-tuning with demonstration setup in LM-BFF [15]. We use $\{3e-6, 4e-6, 5e-6\}$ for the learning rate search of COCO-LM *base++/large++* model, with everything else kept same as described in LM-BFF. With exactly the same pipeline, COCO-LM outperforms RoBERTa under both *base++* and *large++* sizes by significant margins on MNLI-m/mm. Such observations are interesting as COCO-LM's main Transformer does not even see any

| Model | MNLI-m | MNLI-mm |
|---|---|---|
| RoBERTa$_{Base++}$ | 60.1 (1.5) | 61.8 (1.2) |
| COCO-LM$_{Base++}$ | 66.5 (2.1) | 68.0 (2.3) |
| RoBERTa$_{Large++}$ | 70.7 (1.3) | 72.0 (1.2) |
| COCO-LM$_{Large++}$ | 72.0 (1.5) | 73.3 (1.1) |

Table 4: Few-shot prompt-based fine-tuning using RoBERTa and COCO-LM trained on 16 samples per class. Mean (and standard deviation) accuracy results over 5 different splits on MNLI-m/mm are shown.

[MASK] tokens during pretraining but still performs well on predicting masked tokens for prompt-based learning. Note that ELECTRA and COCO-LM variants without the CLM task are not applicable: Their main Transformers are not pretrained by language modeling tasks (thus no language modeling capability is learned to generate prompt label words). This points out the importance, if not necessity, of COCO-LM in the family of ELECTRA-style pretraining models. With the benefits and rapid developments of prompt-based approaches, the lack of language modeling capability is going to limit the potential of ELECTRA's self-supervised learning framework in many real-world scenarios. COCO-LM not only addresses this limitation but also provides better prompt-based learning results.

## 6 Conclusions and Future Work

In this paper, we present COCO-LM, which pretrains language models using Corrective Language Modeling and Sequence Contrastive Learning upon corrupted text sequences. With standard pretraining data and Transformer architectures, COCO-LM improves the accuracy on the GLUE and SQuAD benchmarks, while also being more efficient in utilizing pretraining computing resources and network parameters.

One limitation of this work is that the contrastive pairs are constructed by simple cropping and MLM replacements. Recent studies have shown the effectiveness of advanced data augmentation techniques in fine-tuning language models [16, 38, 51]. A future research direction is to explore better ways to construct contrastive pairs in language model pretraining.

Despite the empirical advantage of this auxiliary-main dual model framework, the auxiliary Transformer training is not influenced by the main Transformer nor learns to generate the optimal pretraining signals for the main model. To better understand and tailor the training of the auxiliary model to the main model is another important future research direction.

## Acknowledgments

We sincerely thank Guolin Ke for discussions and advice on model implementation. We also thank anonymous reviewers for valuable and insightful feedback, especially the suggestion of adding prompt-based fine-tuning experiments.

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
