# A  GLUE Tasks

We provide more details of the tasks included in the GLUE benchmark. Their statistics are listed in Table 5.

**MNLI:** Multi-genre Natural Language Inference [58] contains 393K train examples obtained via crowdsourcing. The task is to predict whether a given premise sentence entails, contradicts or neutral with respect to a given hypothesis sentence.

**QQP:** Question Pairs [48] contains 364K train examples from the Quora question-answering website. The task is to determine whether a pair of questions asked are semantically equivalent.

**QNLI:** Question Natural Language Inference contains 108K train examples derived from the Stanford Question Answering Dataset (SQuAD) [41]. The task is to predict whether a given sentence contains the answer to a given question sentence.

**SST-2:** Stanford Sentiment Treebank [50] contains 67K train examples extracted from movie reviews with human-annotated sentiment scores. The tasks is to determine if the sentence has positive or negative sentiment.

**CoLA:** Corpus of Linguistic Acceptability [57] contains 8.5K train examples from books and journal articles on linguistic theory. The task is to determine whether a given sentence is linguistically acceptable or not.

**RTE:** Recognizing Textual Entailment [2, 10, 21, 17] contains 2.5K train examples from textual entailment challenges. The task is to predict whether a given premise sentence entails a given hypothesis sentence or not.

**MRPC:** Microsoft Research Paraphrase Corpus [12] contains 3.7K train examples from online news sources. The task is to predict whether two sentences are semantically equivalent or not.

**STS-B:** Semantic Textual Similarity [3] contains 5.8K train examples drawn from multiple sources with human annotations on sentence pair semantic similarity. The task is to predict how semantically similar two sentences are on a 1 to 5 scoring scale.

# B  Hyperparameter Settings

Tuning hyperparameter of pretraining is often too costly and we keep most hyperparameters as default. The auxiliary MLM pretraining uses the standard $15\%$ `[MASK]` ratio. The crop transformation in the SCL task uses $10\%$ crop ratio, resulting in a sub-sequence that is $90\%$ long of the original sequence. The softmax temperature in the SCL task is $1$. All pretraining tasks in COCO-LM have equal weights except $\lambda_{\text{copy}} = 50$ since the loss of the binary classification task is much lower than those of the LM tasks, which are over $30,000$-way classification tasks. All token embeddings (used in the input embedding layer and the language modeling head) are shared between the auxiliary Transformer and the main Transformer. The detailed hyperparameters used are listed in Table 6 for pretraining, and Tables 7 and 8 for GLUE and SQuAD fine-tuning, respectively.

All reported methods use exactly the same (or equivalent) set of hyperparameters for pretraining and fine-tuning for fair comparison. For COCO-LM and all the baselines implemented under our setting, all fine-tuning hyperparameters are searched per task; the median results of five runs with the same set of five different random seeds are reported on GLUE and SQuAD.

# C  The Origins of Reported Baseline Scores

The baseline results listed in Table 1 are obtained from their original papers except the following: BERT from Bao et al. [1], RoBERTa *base/base++* GLUE from and SQuAD from Bao et al. [1], ELECTRA *base/base++* GLUE from Xu et al. [61], XLNet *base++* from Bao et al. [1], RoBERTa *base++* SQuAD from Bao et al. [1]. When multiple papers report different scores for the same method, we use the highest of them in our comparisons.

|        | Size  | Task         | Metric(s)         | Domain        |
|--------|-------|--------------|-------------------|---------------|
| MNLI   | 393K  | Inference    | Accuracy          | Misc.         |
| QQP    | 364K  | Similarity   | Accuracy/F1       | Social QA     |
| QNLI   | 108K  | QA/Inference | Accuracy          | Wikipedia     |
| SST-2  | 67K   | Sentiment    | Accuracy          | Movie Reviews |
| CoLA   | 8.5K  | Acceptability| Matthews corr.    | Misc.         |
| RTE    | 2.5K  | Inference    | Accuracy          | Misc.         |
| MRPC   | 3.7K  | Paraphrase   | Accuracy/F1       | News          |
| STS-B  | 5.7K  | Similarity   | Pearson/Spearman. | Misc.         |

Table 5: The list of tasks in GLUE, their training data size, language tasks, evaluation metrics, and domain of corpus.

| Parameters                                   | *base*       | *base++*     | *large++*    |
|----------------------------------------------|--------------|--------------|--------------|
| Max Steps                                    | 125K         | 1.95M        | 1.95M        |
| Peak Learning Rate                           | 5e-4         | 2e-4         | 1e-4         |
| Batch Size                                    | 2048         | 2048         | 2048         |
| Warm-Up Steps                                | 10K          | 10K          | 10K          |
| Sequence Length                              | 512          | 512          | 512          |
| Relative Position Encoding Buckets           | 32           | 64           | 128          |
| Relative Position Encoding Max Distance      | 128          | 128          | 256          |
| Adam $\epsilon$                              | 1e-6         | 1e-6         | 1e-6         |
| Adam $(\beta_1, \beta_2)$                    | (0.9, 0.98)  | (0.9, 0.98)  | (0.9, 0.98)  |
| Clip Norm                                    | 2.0          | 2.0          | 2.0          |
| Dropout                                      | 0.1          | 0.1          | 0.1          |
| Weight Decay                                 | 0.01         | 0.01         | 0.01         |

Table 6: Hyperparameters used in pretraining.

# D   More Implementation Details

**Pretraining and Fine-tuning Costs.** The *pretraining cost* of COCO-LM's CLM task is similar to ELECTRA, which is BERT plus the auxiliary network whose size is $1/3$ of the main network. The addition of SCL task requires one more forward and backward pass on the cropped sequence $X^{\text{crop}}$. With 256 V100 (32 GB Memory), one pretraining run takes about 20 hours in *base* setting, about two-three weeks in *base++* setting, and about three-four weeks in *large++* setting. The *fine-tuning costs* are the same with BERT plus relative positive encodings as the same Transformer model is used.

**MLM Mode for Corrective Language Modeling.** When creating the MLM replaced sequence $X^{\text{MLM}}$, we find it slightly improves the downstream task performance to disable dropout (*i.e.*, set the auxiliary MLM in inference mode) for computing the auxiliary network's output distribution where plausible replacing tokens are sampled. We hypothesize that this leads to more stable generation of challenging replaced tokens to be corrected by the main Transformer and thus improves downstream task results.

**Projection Heads.** For the auxiliary model trained with MLM, we follow the standard MLM head setup in BERT/RoBERTa that includes a linear layer to project the contextualized embeddings from the encoder to same-dimensional vectors before feeding to the final linear layer that outputs the MLM probability. However, we do not include the projection layer for the main model trained with the CLM task (*i.e.*, only having the final linear layer). We find this improves the training stability.

**Masking Special Tokens for Auxiliary Model Training.** BERT only masks real tokens (other than artificial symbols like `[SEP]` and `[CLS]`) for MLM training, while RoBERTa also masks special tokens. We follow the RoBERTa setting which results in slightly improved performance for some tasks.

| Parameters | GLUE Small Tasks Search Space | GLUE Large Tasks Search Space |
|---|---|---|
| Max Epochs | {2, 3, 5, 10} | {2, 3, 5} |
| Peak Learning Rate | *base*/*base++*: {2e-5, 3e-5, 4e-5, 5e-5} *large++*: {7e-6, 1e-5, 2e-5, 3e-5} | *base*/*base++*: {1e-5, 2e-5, 3e-5, 4e-5} *large++*: {5e-6, 7e-6, 1e-5, 2e-5} |
| Batch Size | {16, 32} | 32 |
| Learning Rate Decay | Linear | Linear |
| Warm-Up Proportion | {6%, 10%} | 6% |
| Sequence Length | 512 | 512 |
| Adam $\epsilon$ | 1e-6 | 1e-6 |
| Adam $(\beta_1, \beta_2)$ | (0.9, 0.98) | (0.9, 0.98) |
| Clip Norm | - | - |
| Dropout | 0.1 | 0.1 |
| Weight Decay | 0.01 | 0.01 |

Table 7: Hyperparameter ranges searched for fine-tuning on GLUE. GLUE small tasks include CoLA, RTE, MRPC and STS-B. GLUE large tasks include MNLI, QQP, QNLI and SST-2.

| Parameters | SQuAD Search Space |
|---|---|
| Max Epochs | {2, 3} |
| Peak Learning Rate | *base*/*base++*: {2e-5, 3e-5, 4e-5, 5e-5} *large++*: {7e-6, 1e-5, 2e-5, 3e-5} |
| Batch Size | {16, 32} |
| Learning Rate Decay | Linear |
| Warm-Up Proportion | {6%, 10%} |
| Sequence Length | 512 |
| Adam $\epsilon$ | 1e-6 |
| Adam $(\beta_1, \beta_2)$ | (0.9, 0.98) |
| Clip Norm | - |
| Dropout | 0.1 |
| Weight Decay | 0.01 |

Table 8: Hyperparameter ranges searched for fine-tuning on SQuAD.

# E   More Discussions on PLM Research

Currently, the biggest challenge with PLM research is perhaps its prohibitive computation cost. On one hand, PLMs have influenced a wide range of tasks, and any further technical improvement matters a lot for downstream applications. On the other hand, its expensive computing cost and long experimental cycles pose great challenges for careful and thorough studies of the problem space, as any test of new designs comes with a considerable computing cost—pretraining a new language model can easily consume thousands of dollars, or even millions for extra large models.

Such challenges call for more systematic evaluation pipelines that can accurately and reliably judge whether or not a new PLM is really better than previous ones. Currently, the evaluation of PLMs largely relies on GLUE-style benchmark which contains a set of different tasks that are weighed equally for PLM evaluations—usually the average performance over these tasks is treated as a final measure for the effectiveness of a PLM. However, we find that the small tasks in GLUE have very high variances which may provide unreliable indications for a PLM's performance. For example, on CoLA and RTE, fine-tuning with different random seeds from the same pretrained checkpoint can easily result in a 5-point difference between the best and the worst seed. In contrast, large tasks like MNLI give relatively stable and consistent results for the same model pretrained/fine-tuned with different random seeds, and thus serve as better indicators for PLMs' effectiveness.

In this paper, we try to improve the robustness of our observations, for example, by reporting the downstream performance with different training time for future comparisons under limited computing budget, and also by making our code and models publicly available for the reproducibility of our study. We hope our efforts will facilitate more future research to improve the community's understanding and development of this important problem space.