# OpenReview forum: "COCO-LM: Correcting and Contrasting Text Sequences for Language Model Pretraining"
_NeurIPS.cc/2021/Conference — NeurIPS 2021 Poster_

### Official Review · Reviewer_SaZ9 · 2021-06-29

**Rating:** 7
**Confidence:** 4

**Summary:**

This paper introduces COCO-LM, a pretraining method. The pretraining has two components: *COrrecting*, and *COntrasting*. The COrrecting objective shares most of the efficiency advantages of ELECTRA, while still allowing some language modelling ability. The COntrasting objective ensures that sentences do not all have high cosine similarity, something that occurs in ELECTRA.

* The pretraining method achieves higher accuracy more efficiently than counterparts across various tasks (GLUE, SQuAD 2.0) and settings (base, base with more data and large).
* There is a comprehensive ablation study and analysis section that supports the design choices

**Limitations And Societal Impact:**

This is similar to BERT and has the same potential concerns.

**Main Review:**

This paper introduces COCO-LM, a pretraining method. The pretraining has two components: *COrrecting*, and *COntrasting*.  For correcting, like ELECTRA, the authors first mask some tokens in a sentence. A small auxiliary model is trained to predict those tokens. The main model must then predict which words have been replaced. In contrast with Electra, the main also predicts the original word when it was replaced. For contrasting, the model must predict whether other sentences in a batch are corrupted versions of the sentence (either through cropping or masking) or different sentences altogether.

The COrrecting objective shares most of the efficiency advantages of ELECTRA, while still allowing some language modelling ability. The COntrasting objective ensures that sentences do not all have high cosine similarity, something that occurs in ELECTRA. Instead, sentences will have high cosine similarity if they are similar, which is useful for many downstream tasks, (particularly for e.g: STSB).

The authors compare COCO-LM to other pretraining methods on GLUE and SQuAD 2.0. They experiment with three settings, varying model and data size. Overall, the results are strong. On GLUE, COCO-LM improves by about 1 point on average. On SQuAD 2.0, it improves over existing approaches in the base settings but performs on par with them in the large setting.

In addition, on MNLI the model reaches higher accuracies in less GPU time than other pretraining methods. Given each step is longer, this means that the data efficiency of COCO-LM more than makes up for the overhead the method introduces over e.g ELECTRA.

The ablation study in this paper is very comprehensive and useful. Interestingly, keeping only the ELECTRA setup performs better than the original ELECTRA, likely because of the changes in the auxiliary model setup and due to better optimization (indeed, the authors own implementation of ELECTRA already outperforms the original one). Finally, the authors also convincingly demonstrate the usefulness of their objective design choices.

Overall, this paper makes a compelling case for this new pretraining method. The results are strong, the ablation study comprehensive, the design choices are well motivated and their usefulness is supported by the analysis. Replicating some baselines also shows good scholarship.
With general pretraining methods, it is easy to want more and ask for more experiments (e.g: SuperGLUE), as many tasks and setups are relevant to show the generality of the method, but I feel like the authors case is strong enough in this instance.

The biggest missing piece in the paper is the lack of hyperparameter details, because the appendix is not included. The sections do appear in my pdf’s reader table of content, but not the text. There are also some result concerns, which I detail below.


Pros:
- COCO-LM performs well across various model sizes and tasks and is also more efficient in terms of GPU/h for some level of accuracy.
- The ablation study is comprehensive and compelling. Noteworthy that some design choices show only a small gain (e.g: RTD alone does not perform much worse).
- The analysis of the design choices behind each objective is thorough and convincing.
- The open-sourced model will be useful.

Cons:
- The appendix is currently missing and with it many of the hyperparameter details required for reproducibility.
- Please share at least one run / set of results on the test set for each setting. This is necessary to ensure that the difference is not just in hyperparameter overfitting during fine-tuning.
- I understand that the claim “matches Megatron-3.9Bn, a model 10x bigger on MNLI” is compelling and makes for a great abstract. However, I think it is misleading and poor - scholarship. Indeed, Megatron-3.9Bn is not a great baseline for a 1Bn+ model. For instance, DeBERTA 1.5B+SIFT achieves 92.0/92.1 on MNLI (see Table 12 in their appendix, though not clear whether this is dev or test), quite a bit higher than Megatron 3.9Bn (and COCO-LM).
- Minor: Some typos: Throughout the paper (3 times), please replace data argumentation with data augmentation. “The analyses uses base setting” -> “The analysis uses the base setting” or “The analyses use the base setting”, etc.
- Minor: You can keep only one significant digit for GLUE/SQuAD results. Would make tables easier to read and I am not convinced a difference of <0.1 is significant on those tasks.


~~~ Update author response:

Thanks for the response. Looking forward to the improvements to the paper. Keeping my score.


**Time Spent Reviewing:**

3.5

---

> ### Author Response · Authors · 2021-08-10
> **Response to Reviewer SaZ9**
>
> Thank you for your review. Per reviewers’ suggestions we have conducted several new experiments and analyses, discussed in the general response. Here we discuss in more detail on your concerns and how we plan to improve them in our next version.
>
> * Based on our understanding of the submission format, we put the appendix in the “Supplementary Material” part of the submission instead of the main PDF file. You can find the appendix by unzipping the supplementary material under the “Abstract” field. To sum up, we try our best to follow the same hyperparameter settings in pretraining and fine-tuning with existing research. There are inevitable differences, e.g., in the versions of pretraining corpus, preprocessing, fine-tuning pipeline, etc. Thus, we also train RoBERTa base and ELECTRA base ourselves with everything else equal for more direct comparisons.
>
> * We include the testing results of COCO-LM base and base++ in the general response from our private submissions to the GLUE leaderboard. These confirm that our gains observed on Dev also hold on hidden test.
>
> * We agree with your suggestion. We will make more careful statements following the discussion in the general response (first table).  Note that all our comparisons are conducted in the single-task fine-tuning setting, a widely used setting to directly evaluate pretraining models’ performance on the Dev set,, e.g., in BERT, RoBERTa, and ELECTRA. DeBERTA 1.5B +SIFT used the SIFT fine-tuning method which is very effective on GLUE leaderboard. In addition, their reported Dev numbers on RTE, MRPC, and STS-B (https://github.com/microsoft/DeBERTa/) are fine-tuned from MNLI checkpoints which is an effective technique when making GLUE leaderboard runs. We fine-tuned DeBERTa V2-XXLarge on these three tasks using their released checkpoint with our GLUE pipeline in the single task setting. The results are listed in the general response for reference.
>
> * We will fix these typos and use single digit for results. We will conduct more proofread in the next version.

---

### Official Review · Reviewer_cYxo · 2021-07-12

**Rating:** 6
**Confidence:** 4

**Summary:**

This paper tackles two challenges in current self-supervised learning frameworks, including the pre-training efficiency and the anisotropy of text representations. To address these two challenges, the authors propose the COCO-LM framework that pre-trains language models by correcting and contrasting corrupted text sequences. COCO-LM follows ELECTRA-style pre-training such that an auxiliary language model is employed to corrupt text sequences, upon which two new tasks are constructed to pre-train the main model. The first task is Corrective Language Modeling (CLM) to detect and correct tokens replaced by the auxiliary model, in order to better capture token-level semantics. The second task is Sequence Contrastive Learning (SCL) to align text sequences originated from the same source input while ensuring uniformity in the representation space. Experimental results on GLEU and SQuAD demonstrate that COCO-LM can improve the performance of pre-trained models and also the efficiency of pre-training.

**Limitations And Societal Impact:**

There is no negative societal impact.

**Main Review:**

Strengths
1.	The problems tackled in this paper are critical. While ELECTRA-style improves the pre-training efficiency and achieves improvements on language understanding tasks, the language modeling capability is compromised due to the replaced token detection (RTD) task. This paper incorporates both RTD and LM objectives to fully utilize the self-supervision signals in the text. Besides, representations by BERT-style or ELECTRA-style pre-trained models are often facing the degeneration issue, which may require sophisticated post-adjustment to achieve good performance in fine-tuning.
2.	Experiments on various language understanding tasks demonstrate that COCO-LM can achieve the performance of SOTA approaches with 50% of the pre-training GPU hours. With the same training time, COCO-LM outperforms other approaches with non-trivial margins.


Weaknesses
1.	While the paper claim the importance of language modeling capability of pre-trained models, the authors did not conduct experments on generation tasks that are more likely to require a well-performing language model. Experiments on word similarity and SquAD in section 5.3 cannot really reflect the capability of language modeling. The authors may consider to include tasks like language modeling, machine translation or text sumarization to strenghen this part, as this is one of the main motivations of COCO-LM.
2.	Analysis of SCL in section 5.2 regarding few-shot abaility looks not convincing. The paper claims that a more regularized representation space by SCL may result in better generalization ability in few-shot scenarios. However, results in Figure 7(c) and (d) do not meet our expectation such that COCO-LM achieves much more improvements with less labels and the improvements will gradually disappear with more labels. Besides, the authors may check if COCO-LM brings benefits to sentence retrieval tasks with the learned anisotropy text representations.
3.	The comparison with Megatron is a little overrated. The performance of Megatron and COCO-LM is close to other approaches, for examples, RoBERTa, ELECTRA, and DeBERTa, which are with similar sizes as COCO-LM. If the author claim that COCO-LM is parameter-efficient, the conclusion is also applicable to the above related works.


Questions for the Authors
1.	In experimental setup, why did the authors switch the types of BPE vocabulary, i.e., uncased and cased. Will the change of BPE cause the variance of performance?
2.	In Table 2, it looks like COCO-LM especially affects the performance on CoLA and RTE hence the final performance. Can the authors provide some explanation on how the proposed pre-training tasks affect the two different GLEU tasks?
3.	In section 5.1, the authors say that the benefits of the stop gradient operation are more on stability. What stability, the training process? If so, are there any learning curves of COCO-LM with and without stop gradient during pre-training to support this claim?
4.	In section 5.2, the term “Data Argumentation” seems wrong. Did the authors mean data augmentation?


Typos
1.	Check the term “Argumentation” in line 164, 252, and 314.
2.	Line 283, “a unbalanced task”, should be “an unbalanced task”.
3.	Line 326, “contrast pairs”, should be “contrastive pairs” to be consistent throughout the paper?


**Time Spent Reviewing:**

6

---

> ### Author Response · Authors · 2021-08-10
> **Response to Reviewer cYxo**
>
> Thank you for your review. Following your comments, we have conducted several new experiments and analyses, as listed in the general response. Here we discuss in more detail regarding the weaknesses and questions you pointed out.
>
> Regarding the weaknesses:
> 1.	We agree that the word similarity test and SQuAD are indirect reflections of COCO-LM’s language modeling benefits over ELECTRA. The prompt-based approaches suggested by reviewer gYa8, which rely on the pretrained model’s language modeling head to generate label words with only a handful of training samples, is a much more direct evaluation. As shown in the general response, COCO-LM works well in this setting, while without CLM neither COCO-LM variants nor ELECTRA are directly applicable for prompt-based approaches. This confirms the benefits of COCO-LM’s language modeling capability in the ELECTRA-style pretraining family. We also view this prompt-based approach as a form of generation task as the model "generates" the labeling words. The traditional NLG tasks like language modeling, MT, and summarization are more suitable for evaluating auto-regressive pretrained models or encoder-decoder-based models. To evaluate the bidirectional encoder-only models, e.g., BERT, ELECTRA, and COCO-LM in these NLG tasks, one possible way is to initialize an encoder-decoder model (e.g., BART) with the pretrained encoder-only model and training the decoder from scratch. However, that is a less direct evaluation. We hope our results with prompt-based learning have sufficiently demonstrated the importance of the language modeling capability of pre-trained models.
>
> 2.	As discussed in the paper, SCL provides a form of regularization in the sequence space. Its benefits are not as obvious as CLM which enables the language modeling capability.  That said, we observe stable improvements from SCL on downstream tasks with limited fine-tuning labels. The improvements of using SCL in Fig 7(c) and (d) versus not using SCL are 0.7 in MNLI-m/mm in average accuracy when only using 1% and 5% training labels. The absolute gain is similar to the margin between DeBERTa (base) over ELECTRA (base), to put into perspective. The benefit of SCL’s regularization also leads to better performance on GLUE’s smaller tasks including RTE and MRPC. (Single task CoLA gains are not very meaningful as it is quite unstable). In Table 2, COCO-LM base and COCO-LM (SCL+RTD) outperform their no SCL counterparts, COCO-LM (CLM Only) and COCO-LM (RTD Only) by significant margins on RTE and MRPC. We will improve the discussion of SCL’s benefits and clarify of the tables/figures in the next version.
>
>
> 3.	We will improve the discussion of COCO-LM’s parameter efficiency, with more careful statements and added comparisons with DeBERTa’s XL and XXL models in the general response. Though Megaton-3.9B may not be as effective as newer models, it is one of the largest pretrained BERT-style encoders and has decent empirical gains (1.2) over RoBERTa Large++ on MNLI, while COCO-LM Large++ is on par with Megatron-3.9B on MNLI. In comparison with DeBERTa XL(++) and XXL(++), which is the recent SOTA in beyond-Large size encoders, COCO-LM Large++ is more parameter efficient as it provides comparable performances with DeBERTa XL variations which use more than double parameters and more sophisticated transformer architectures. We will include these new comparisons and be more clear in our statements in our next version.
>
> Regarding your detailed questions:
> 1.	We use one BPE vocabulary for WikiBook (base) and the other one for the full 160G pretraining data (base++ and large++) mainly to be as close with the baseline settings as possible. Previous work using WikiBook often choose 32K uncased (BERT base and ELECTRA base) and those on the full 160G data often use 64K cased (RoBERTa base++/large++ and UniLM). Our observation is that, sometimes, the uncased vocab works slightly better on GLUE and slightly worse on SQuAD2.0, but the differences are minor and within random range. Our choice of vocabulary is solely for consistency with previous work, not for empirical effectiveness.
>
> 2.	Following the discussion in response to your weakness #2, except on CoLA which is known to be non-stable in single-task fine-tuning, we found COCO-LM more effective on tasks with limited training labels, including few-shot MNLI, RTE, and MRPC. This reflects the benefit of SCL’s regularization. The stability on smaller tasks is also observed in larger models, as shown in our general response.
>
> 3.	The stability benefit of stop gradient is on training convergence. When trained without stop gradient, we have encountered several runs that the CLM task may make the training diverge. We will make this clear in the next version.
>
> 4.	Yes, it should be data augmentation. Thanks for spotting these typos and we will do more proofreads.

---

### Official Review · Reviewer_gYa8 · 2021-07-16

**Rating:** 7
**Confidence:** 4

**Summary:**

The paper proposes a pre-trained technique based on correcting tokens and contrasting sequences of text. The correcting objective is based on ELECTRA with a modification in the formulation of multi-task setup where the one task is to identify the corrupted tokens and the other task is to correct it. The contrasting objective aims to maximize the similarity between similar sentences while pushing dissimilar sentences away in cosine space. With the two modifications, COCO-LM is trained much faster than ELECTRA. The paper also presents a comprehensive evaluation on GLUE and SQUAD 2.0 benchmarks and ablation study of their method.

**Limitations And Societal Impact:**

The authors have adequately addressed the limitations and potential negative societal impact of their work.

**Main Review:**

- In terms of originality, COCO-LM is built on ELECTRA, thus it lacks a novelty aspect. ELECTRA does not have a sequence-level objective, COCO-LM uses a contrasting objective (SCL) for sequences thus it performs better than ELECTRA on sequence level tasks (i.e. GLUE). As can be seen in table 2, SCL + RTD improves the average score on GLUE significantly compared to RTD (i.e., ELECTRA). Having said that, I think the contribution of the paper is quite marginal, however I think its strong point is the efficiency in training time.

- The paper is well-presented and contains a comprehensive list of experiments showing the effect of each of the modified components. What I find missing is that the paper hasn’t addressed the claim about missing language modelling benefit (line 107-113). Although the authors presented word-similarity (table 3, line 292) results, I think that it’s rather a proxy evaluation but is not convincing enough about the benefit of language modeling capabilities. As the prompt-based approach [1,2,3] is gaining popularity recently, I wonder if COCO-LM can be used in this setup where the model is required to generate (i.e., assessing language modeling capability). This is something MLM such as RoBERTA, BERT are doing quite well.

**post rebuttal**: As authors run additional experiments with prompt tuning and showed good results. I increase my score to 7,

[1] Tianyu Gao, Adam Fisch, and Danqi Chen. Making pre-trained language models better few-shot learners.
[2] Xiao Liu, Yanan Zheng, Zhengxiao Du, Ming Ding, Yujie Qian, Zhilin Yang, and Jie Tang. GPT understands, too.
[3] Guanghui Qin and Jason Eisner. Learning how to ask: Querying lms with mixtures of soft prompts.


**Time Spent Reviewing:**

2.5

---

> ### Author Response · Authors · 2021-08-10
> **Response to Reviewer gYa8**
>
> Following your suggestion, we conduct prompt-based fine-tuning experiments for our base models. The results are listed in the end of our general response. The benefits of better language modeling capability from COCO-LM are more directly observed in this setting.
> We sincerely thank you for pointing out prompt-based approach as a potential benefit of COCO-LM’s better language modeling capability. In our next version, we will include this prompt-based study and acknowledge that it is your suggestion.

---

### Official Review · Reviewer_uxzq · 2021-07-16

**Rating:** 7
**Confidence:** 5

**Summary:**

Authors propose a pretrained language model that extends Electra. They operate within the generator/discriminator proposed by Electra. The key contributions in the paper are a couple of novel loss functions that allow the discriminator to predict the tokens rather than binary labels and a contrastive loss for the sequence embedding. With a detailed analysis they show that the resulting token level and sentence embedding are much better at discriminating random input. They also demonstrate that this improves the quality on the GLUE and SQUAD datasets. They further demonstrate that their approach results in faster training and improved parameter efficiency.

**Ethical Concerns:**

None.

**Main Review:**

Positives

- The results that authors demonstrate are very compelling and reliable.
- The analysis on the token and sentence level embedding is insightful.

Negatives

- Their approach is an extension of Electra rather than something fundamentally new.
- The exposition could be improved.

Detailed comments

- Related works section could be more elaborate.
- It would also be beneficial if the authors included a brief paragraph or two on how this work can be extended to the multilingual setting.
- All-Token MLM is just referred to in line 170. Since this is critical for understanding the paper readers could be eased into it rather than assuming they have read ELECTRA.
- In figure 3 the author shows training time in GPU hours. It is good to plot against examples seen or number of training steps (assuming all three used the same batch size). This would help clarify that there are no inefficiencies in the implementation of Roberta/ELECTRA.
- Line 249, authors mention that the benefit of stop gradients are for stability. It is not clear what they refer to. Removing stop gradients seems to have a very positive effect on the average scores.


**Time Spent Reviewing:**

3

---

> ### Author Response · Authors · 2021-08-10
> **Response to Reviewer uxzq**
>
> Thanks for your comments. We agree with your suggestions will revise our paper accordingly.
> Besides the added experimental results and analyses in the general response, here are more detailed discussions regarding your comments.
> * We will allocate more space in the related work section in the next version. We will greatly appreciate if you could suggest potentially missed/new related work.
> * We will add discussions in the future work section on how COCO-LM can be extended to the multi-lingual setting. In general, the data efficiency of ELECTRA-style pretraining makes it quite effective in cross-lingual pretraining. The SCL task also naturally fits with the parallel corpus in some multi-lingual pretraining setting. In fact, there have been recent parallel work that explored these properties in the multi-lingual setup. We will leave these references and discussion in the next version to keep anonymity.
> * We will add the details of All-Token MLM in Sec 3.1 in the next version.
> * Our RoBERTa and ELECTRA implementations share most codes with COCO-LM. We are confident their implementations are at the same efficiency and their experiments used exact same computing environments and GPU setups.  All the three models in Figure 3 are pretrained for the same 125K steps (of 2048 effective batch size). Thus to plot RoBERTa/ELECTRA against the number of training steps/examples would move them to the very end (right most) of the x-axes in Figure 3. That will make COCO-LM look better but it is not a fair comparison.  We chose GPU hours which is a better indicator of training effectiveness.
> * The stability benefit of stop gradients is to reduce training divergence. Removing it sometimes leads to better overall results, but the model’s gradient norms are less stable and sometimes the model training diverges.

---

### Author Response · Authors · 2021-08-10
**General Response and Updated Analyses per Reviewers' Suggestions**

We thank all reviewers for their thoughtful discussions and suggestions. Based on reviewers’ suggestions, we provide the following new experimental results and analyses:

* Updated COCO-LM Large++ results with the same pretraining steps of RoBERTa Large++ and comparing with DeBERTa’s variations of larger sizes.

* GLUE Test results of COCO-LM base/base++.

* Analyses of COCO-LM’s language modeling capabilities in the prompt-based approaches (LM-BFF [1]) for few-shot learning


[1] Tianyu Gao, Adam Fisch, and Danqi Chen. Making pre-trained language models better few-shot learners. https://github.com/princeton-nlp/LM-BFF


### More detailed results and discussions:

**Large Model Performances.** The following table lists our updated COCO-LM Large++ performance on GLUE and SQuAD 2.0 development setting. The update is that we pretrained COCO-LM Large++ with the same pretraining steps as RoBERTa Large++ (4 billion samples grouped in 2048 batches) for a more fair comparison. We further examine COCO-LM’s parameter efficiency by comparing COCO-LM large with DeBERTA XL and XXL variations. DeBERTA XL results are obtained from https://github.com/microsoft/DeBERTa/ while XXL is fine-tuned by ourselves using our fine-tuning pipeline, to obtain signal task results on RTE, MRPC and STS-B. All numbers are in the single-task setting without per-task tricks nor sophisticated fine-tuning techniques.


| Large++                | MNLI | QNLI |  QQP | SST-2 | CoLA | RTE  | MRPC | STS-B | GLUE Average | SQuAD 2.0 EM | SQuAD 2.0 F1 |   |
|:------------------------|:----:|:----:|:----:|:-----:|------|------|------|-------|--------------|--------------|--------------|---|
| DeBERTA   XL (750M)    | 91.4 | n.a. | n.a. |   97  | n.a. | n.a. | n.a. | n.a.  | n.a.         | n.a.         | n.a.         |   |
| DeBERTA V2 XL (900M)     | 91.5 | 95.8 | 92.3 |  97.5 | 71.1 | n.a. | n.a. | n.a.  | n.a.         | 88.9         | 91.4         |   |
| DeBERTA V2 XXL (1.5B)  | 91.7 | 96.1 | 92.6 |  96.9 | 70.1 | 87.3 | 91.4 | 91.4  | 89.7         | 89.7         | 92.2         |   |
| COCO-LM Large++ (367M) | 91.5 | 95.7 | 92.8 |  96.9 | 73.9 | 91.0 | 92.2 | 92.7  | 90.8         | 88.2         | 91.0         |   |


With the same pretraining steps as RoBERTa Large, COCO-LM now outperforms all baselines in table 1 on SQuAD 2.0 as well: 88.2/91.0 versus previous best in Large++ setting, DeBERTa’s 88.0/90.7.

With less than half of parameters, COCO-LM outperforms both DeBERTA XL versions on MNLI, despite DeBERTA using a more sophisticated transformer architecture than COCO-LM’s basic BERT + relative position embedding.  COCO-LM Large++ is slightly behind DeBERTA V2 XXL using about 25% parameters on larger tasks but performed better on tasks with fewer labels, including RTE, MRPC and STS-B, yielding a better GLUE average in the single task fine-tuning setting.

**GLUE Test.** The following table lists the GLUE test results by making private submissions to the leaderboard. For reference, we list ELECTRA’s test numbers from Table 8 of the original ELECTRA paper. We used the same approach as in ELECTRA to pick the best runs on GLUE Dev to submit to test. All results are in the single task vanilla fine-tuning, no ensembling, task-specific tricks nor sophisticated fine-tuning techniques.

| GLUE   Test Single Task | MNLI |  QQP | QNLI |  SST | CoLA | RTE  | MRPC | STS  | GLUE Average |   |   |   |
|:------------------------|:----:|:----:|:----:|:----:|------|------|------|------|--------------|---|---|---|
| ELECTRA base      | 85.8 | 89.1 | 92.7 | 93.4 | 59.7 | 73.1 | 86.7 | 87.7 | 83.5         |   |   |   |
| COCO-LM base      | 88.2 | 89.9 | 93.3 | 94.9 | 61.9 | 81.5 | 87.8 | 88.6 | 85.8         |   |   |   |
|     ELECTRA base ++     | 88.5 | 89.5 | 93.1 | 96.0 | 64.6 | 75.2 | 88.1 | 90.2 | 85.7         |   |   |   |
|     COCO-LM base ++     | 89.4 | 89.9 | 93.5 | 95.4 | 61.4 | 80.8 | 88.4 | 90.4 | 86.2         |   |   |   |


The empirical gains of COCO-LM on GLUE Dev hold on the hidden test set. We have used roughly the same hyperparameter search space on GLUE Dev as in ELECTRA and RoBERTa. More extensive hyperparameter search, e.g., using the search space of DeBERTa, may further improve the GLUE Dev results.

**Few-Shot Prompt-Based Fine-Tuning Results.** The following table includes the prompt-based fine-tuning experiments for our base/large models, using the LM-BFF open-source code [1], following the same few-shot manual prompt fine-tuning + demonstration setting in the original paper. The numbers in the table are mean (and standard deviation) accuracy on MNLI-m/MNLI-mm.

|       Few-Shot Fine-Tuning with LM-BFF      |   MNLI-m   |   MNLI-mm  |
|:-----------------------------------|:----------:|:----------:|
|     RoBERTa base (Ours)         | 50.7 (2.9) | 52.5 (3.2) |
|          COCO-LM base             | 59.5 (1.6) | 61.7 (1.7) |
|     RoBERTa large (reported by LM-BFF paper) | 70.7 (1.3) | 72.0 (1.2) |
|     COCO-LM large | 72.0 (1.5) | 73.3 (1.1) |

With the exact same pipeline, COCO-LM base/large outperformed RoBERTa base/large by significant margins on MNLI-m/mm with LM-BFF. Note that ELECTRA and COCO-LM variants without the CLM pretraining task are not applicable: Their main Transformers are not pretrained by language modeling tasks (thus no language modeling heads are available to generate prompt label words).
This points out the importance, if not necessity, of COCO-LM in the family of ELECTRA-style pretraining models. With the benefits and rapid developments of prompt-based approaches, the lack of language modeling capability is going to limit the benefit of ELECTRA’s self-supervised learning framework in many real-world scenarios. COCO-LM not only addresses this limitation but also provides better prompt-based effectiveness.

---

### Decision · Program_Chairs · 2021-09-27

**Decision:**

Accept (Poster)

**Comment:**

This work improves ELECTRA language pretraining approach by introducing Corrective Language Model task (so the model can generate words) and Sequence Contrastive Learning (so sentence representations are more informative). I agree with the reviewers that the authors have conducted extensive experiments to show that the approach is effective and the analyses are also very thorough and insightful, e.g., analyzing cosine similarities of sentence vectors. There are a few points that the original paper wasn't convincing, e.g., Coco-LM wasn't better than ELECTRA in Large++ setup and only word-similarity is used to demonstrate the main point that Coco-LM has better language modeling capabilities. The new results added for the Large++ and Few-Shot Prompt-Based Fine-Tuning Results convincingly addressed these concerns; hence, I recommend Accept.